# Development of an Artificial Soft Solid Gel Using Gelatin Material for High-Quality Ultrasound Diagnosis

**DOI:** 10.3390/diagnostics14030335

**Published:** 2024-02-04

**Authors:** Minchan Kim, Kicheol Yoon, Sangyun Lee, Mi-Seung Shin, Kwang Gi Kim

**Affiliations:** 1Medical Devices R&D Center, Gachon University Gil Medical Center, 21, 774 Beon-gil, Namdong-daero, Namdong-gu, Incheon 21565, Republic of Korea; kormd98@naver.com (M.K.); kcyoon98@gachon.ac.kr (K.Y.); l0421h@gmail.com (S.L.); 2Premedicine Course, College of Medicine, Gachon University, 38-13, 3 Beon-gil, Dokjom-ro 3, Namdong-gu, Incheon 21565, Republic of Korea; 3Department of Health and Safety Convergence Sciences & Health and Environmental Convergence Sciences, Korea University, 145, Anam-ro, Seongbuk-gu, Seoul 02841, Republic of Korea; 4Division of Cardiology, Department of Internal Medicine, Gil Medical Center, Gachon University College of Medicine, 21 Namdong-daero 774 Beon-gil, Namdong-gu, Incheon 21565, Republic of Korea; 5Department of Biomedical Engineering, College of Health Science, Gachon University, 191 Hambak-moero, Yeonsu-gu, Incheon 21936, Republic of Korea; 6Department of Health Sciences and Technology, Gachon Advanced Institute for Health Sciences and Technology (GAIHST), Gachon University, 38-13, 3 Beon-gil, Dokjom-ro, Namdong-gu, Incheon 21565, Republic of Korea

**Keywords:** ultrasound test soft solid gel (ATS-539), ultrasound gel, gelatin powder, sector probe, monitoring

## Abstract

For ultrasound diagnosis, a gel is applied to the skin. Ultrasound gel serves to block air exposure and match impedance between the skin and the probe, enhancing imaging efficiency. However, if use of the ultrasound gel exceeds a certain period of time, it may dry out and be exposed to air, causing impedance mismatch and reducing imaging resolution. In such cases, the use of a soft, solid gel proves advantageous, as it can be employed for an extended period without succumbing to the drying phenomenon and can be reused after disinfection. Its soft consistency ensures excellent skin adhesion. Our soft solid gel demonstrated approximately 1.2 times better performance than water, silicone, and traditional ultrasound gels. When comparing the dimensions of grayscale, dead zone, vertical, and horizontal regions, the measurements for the traditional ultrasound gel were 93.79 mm, 45.32 mm, 103.13 mm, 83.86 mm, and 83.86 mm, respectively. In contrast, the proposed soft solid gel exhibited dimensions of 105.64 mm, 34.48 mm, 141.1 mm, and 102.8 mm.

## 1. Introduction

Gels are used for treatment and diagnosis via medical ultrasound. In particular, gels are used by applying them to the skin [1]. Ultrasound gel is used to match the characteristic impedance of the probe (transducer) and the skin. The reason for this is that the mechanical vibration energy of ultrasonic waves is transmitted through air to the skin. When air is exposed between the probe and the skin, the characteristic impedance becomes mismatched, so the efficiency of ultrasonic energy transfer between the probe and the skin is greatly reduced [2,3]. Therefore, image quality and treatment effectiveness may be reduced [4,5,6,7]. Thus, the use of ultrasound gel plays a very important role in preventing air exposure between the probe and the skin and matching the characteristic impedance [8].

During the process of using ultrasound gel, patients may complain of discomfort due to the gel’s cold feeling on the skin and foreign body sensation [7]. Additionally, the limitation of using the traditional gel is that if the ultrasound exceeds approximately 10 to 15 min, the gel will dry naturally. Therefore, as the gel dries, imaging results may decline due to air exposure between the probe and the skin and mismatch in characteristic impedance [5,9,10], which cause image resolution to decrease and diagnostic efficiency to be reduced [11].

To solve the problem of gel drying, methods such as the use of water and silicon phantoms have been studied [12,13,14,15]. The use of water phantoms in medical ultrasound has been reported to have positive evaluations of high safety [12,13,14,15]. However, during the ultrasonic energy transfer process, problems such as scattering, absorption, heterogeneity, backscattering, calibration difficulties, and setup complexity may be encountered. These interference problems may reduce echo signals and thus reduce diagnostic visibility. In particular, water can exhibit enhanced scattering with increasing tissue depth due to its strong scattering properties associated with the absorption and reflection of photons [12,13]. As a result, image clarity may be compromised during ultrasound penetration [12]. Additionally, in a clinical environment, various characteristics of the density of tissues or materials can be an obstacle to transmitting ultrasonic energy. Setting up and configuring a water phantom experiment can be complex and reproducibility can be difficult depending on the experimental environment or equipment [12]. Therefore, reproducibility of gels used in ultrasound diagnosis is very important.

For silicon applications, acoustic properties, photoacoustic generation, backscattering, homogeneity, processing complexity, compatibility, cost, and availability are important [12,14,15]. However, the speed and reflection (echo) characteristics of sound waves may be adversely affected due to impedance differences between the ultrasound probe, tissue, and silicone material [13]. Therefore, to overcome these problems, this paper proposes a method of producing a soft solid gel that can be used for a long time by enabling reproducibility within the range where the gel is not dried. A soft solid gel was produced using a gelatin component that has impedance similar to that of human skin, and imaging tests and analysis were performed by using a test phantom (ATS-539) to obtain the results.

## 2. Manufacturing Analysis and Methods

### Analysis of the Ultrasonic Gel Characteristics

Diagnostic ultrasound gel is used to prevent air exposure between the probe and the skin and provide high clarity and efficiency for images through impedance matching, as shown in Figure 1. Therefore, gel is essential for accurate diagnosis. However, the gel dries if its use exceeds a certain period of time. Therefore, to overcome these problems, it is proposed to use a padtype gel to replace liquid gel.

The design of a pad-type gel using gelatin is innovative, and the soft solid pad is proposed to replace liquid gel, as shown in Figure 2. The proposed method seeks to obtain results using gelatin. For this purpose, the gel production method and simulation method to prove the results were analyzed, and the ATS-539 test phantom was used to obtain the results. This study evaluated the performance of the proposed gel pad through video-ultrasound virtual diagnosis and sought to prove the superiority of the results. To achieve this goal, the overall flow process for research and obtaining results was introduced, as shown in Figure 2.

Figure 2 shows the manufacturing process and the apparatus used for soft solid gel preparation. The time required for the proposed soft solid gel’s preparation is also indicated. The figure shows the production process of edible gelatin (Dongsung Kit Co., Ltd., Seoul, Republic of Korea) by melting it using a hot plate. An infrared (IR) thermometer (UT302C thermometer @ Unit) was employed to monitor and record the temperature during the soft solid gel preparation. A 50 mL beaker was used to mix the gelatin and saline solution, with stirring accomplished using a 30 cm horizontal bar. In addition, a frame measuring 9.5 (W) × 21.5 (L) cm × 1 cm (h) was prepared to hold a Falcon tube (45 mL) containing the gelatin liquid, as shown in Figure 3. This was stored in a refrigerator.

The gelatin mixture was prepared by combining gelatin and liquid (water or normal saline solution) in a specific ratio, typically 1:2, according to the stoichiometric ratio represented by Equation (1), where x and y are gelatin and water, respectively. This equation aims to achieve a soft solid gel by preparing the mixture in such proportions, making it possible to create a soft solid gel with human skin-like characteristics [16]. The mixture was heated on a heating plate at around 70 °C to dissolve the gelatin completely. Continuous stirring ensured thorough mixing, as depicted in Figure 2. The resulting clear solution was then poured into the prepared mold or container to determine the shape and size of the soft solid gel, as shown in Figure 4.

After solidification, the soft solid gel was refrigerated for approximately 24 h to maintain its shape; the formula is given in Equation (1) [17,18].
x:y = C_6_H_12_O_6_:H_2_O = 1:2(1)

The weight of the soft solid gel was 450 g. The sample was prepared to create an ultrasound soft solid gel with a capacity of 180 g. The method of producing gelatin using a 250 mL glass beaker was applied, which is shown in Figure 5.

Figure 6 shows the 1000 cc (b1) and 50 cc (b2) stainless steel beakers that were placed on a hot plate. Subsequently, a 250 mL glass beaker was used, and the gelatin powder was boiled at around 70 °C for approximately 24 min. This method was the most optimal within the temperature range at which gelatin can effectively melt [19]. To achieve the desired thickness (6.10 mm) and elasticity (±13.07), the value was determined using Equation (2) [20].

Gelatin undergoes chemical changes. It can maintain elasticity due to the elastin reaction force. Here, F and −k represent the force and gelatin constants, respectively, and x represents the displacement. The gelatin was manufactured in 20 g, 30 g, 40 g, and 50 g samples, respectively, as shown in Equation (2).
(2)F=−k × Δx

Gelatin in the 20 g sample had low elasticity. Therefore, the gelatin was cut, and the elasticity of samples weighing 22 g, 24 g, 26 g, 28 g, and 30 g was measured. The experimental results showed that the 22 g sample demonstrated good elasticity. The gelatin was stirred for approximately 10 min until it was completely dissolved. Afterward, the gelatin solution was poured into a container to create the soft solid gel, and the solution was refrigerated in the container for 24 h. The 1:2 ratio between the gelatin sample and the liquid in the syringe was the most crucial aspect in the fabrication of the soft solid gel [21].

When applying the manufactured soft solid gel for ultrasound imaging, careful observation was necessary. The ultrasound image evaluation experiment consisted of the following five areas: dead zone, vertical and horizontal measurement calibration, axial and lateral resolution, grayscale, and sensitivity. Evaluation of these measurement areas ensured the accuracy of the produced soft solid gel [22].

## 3. Results

In the present study, a conventional ultrasonic gel, the proposed soft solid gel, a probe (with an output frequency of 3.5 MHz) [23], and an ultrasonic evaluation soft solid gel (ATS-539) were used for ultrasound examination. The ATS-539 is typically used for ultrasonic testing. Experiments were conducted to compare the performance of the conventional ultrasonic gel and the proposed soft solid gel, as shown in Figure 7.

To obtain the experimental results, the conventional ultrasonic gel was applied to the ATS-539 for conventional ultrasonic evaluation, and video evaluation was performed using a probe. Simultaneously, the proposed soft solid gel was placed on the ATS-539, and imaging evaluation was performed using a probe. Ultrasonic gel ensures effective transmission of ultrasound waves and good contact with the skin or the subject under examination. Ultrasonic gel must have suitable acoustic properties, including sound speed and impedance, for efficient ultrasound wave transmission and reception. It should be non-toxic for specific applications in the human body and enhance tissue visualization in ultrasound imaging. Considering these factors, a soft solid gel is considered superior [24]. When comparing the soft solid gel produced with the conventional ultrasound gel, it was observed during a 15 min video evaluation using the ultrasound gel that heat was generated on the ultrasound probe (temperature of 34 °C), causing drying of 22% of the gelatin mass (22 g), with a dry mass of 5 g as given by Equation (3) (temperature of 34 °C), as shown in Figure 8 [22].
(3)$$M=\frac{dry\;mass}{gelatin\;mass}\times 100\%$$

Figure 9a–d show the comparative evaluation of performance when using ATS-539 with different types of ultrasound gels, including the proposed gelatin gel, water-based soft solid gel, and silicon-based soft solid gel. In Figure 9a, the imaging results show the ATS-539 used with a probe coated with ultrasound gel. Figure 9b presents the results obtained from testing the soft solid gel fabricated for this experiment. Furthermore, Figure 9c,d exhibit the imaging results obtained from testing commercially available water-based soft solid gel and silicon-based soft solid gel, respectively.

For the proposed soft solid gel, as depicted in Table 1, the values for grayscale, dead zone, vertical zone, and horizontal zone are as follows. When comparing with other manufactured soft solid gels, we considered parameters such as grayscale, dead zone, vertical zone, and horizontal zone. In addition, we included the standard deviation of ultrasound gel, soft solid gel, water soft solid gel, and silicon soft solid gel, as given by Table 1. The results, including numerical values, along with the corresponding table, average values, and standard deviations, were compared. From the comparison of each parameter (grayscale, dead zone, vertical zone, horizontal zone), it is evident that the dead zone was relatively lower, while the grayscale, vertical zone, and horizontal zone exhibited higher average values.

## 4. Discussion

To obtain the elastin ratio corresponding to the conventional gel and gelatin material, the moisture rate (M) was calculated using Equation (3) [25]. The conventional gel was formulated from propylene glycol (CH_3_ CH COH CH_2_ OH) with a density of 1.04 g/cm^3^, while the chemical formula of the gelatin was a combination of collagen, alkali, and protein (C_6_H_12_O_6_), and its density was 1.2 g/cm^3^, respectively. The gelatin was higher in elastin than conventional gel due to its high density [25].

The thermal conductivity values of the conventional gel and gelatin powder were 0.34 W/m-K and 0.55 W/m-K, respectively, with a temperature range of 70 °C to 90 °C at 50% H_2_O [25,26]. Ultrasonic reflection (echo) is crucial for high-resolution monitoring [27]. To evaluate the performance of the proposed method, the first experimental approach involved conducting experiments based on physical simulations and numerical results to substantiate the findings. The second method utilized the ATS-539 phantom, commonly used for performance testing, to demonstrate the excellence of the experimental results through image evaluations.

The proof was established by analyzing the amount of ultrasound entering the gelatin material, the ratio of reflected quantity for monitoring, and the efficiency of input and output power based on Equations (4)–(11), as proposed. For this analysis, the gelatin material underwent a transformation into a rubber-like form. Consequently, there was a need to reevaluate the acoustic wave velocity (*c*) and acoustic impedance (*Z*) of the rubber-like material (gelatin). The characteristic impedance (*Z_T_*) of the transducer, elastic modulus (*E*), and density (*ρ*) of the gelatin material were determined as 30.0 MRay1, 14.26 MPa, and 1.196 g/cm^3^, respectively [28,29,30,31]. Here, elastic modulus (*E*) refers to the ratio of the cross-change rate of transverse and longitudinal waves when ultrasonic waves mechanically vibrate in the transducer [32]. At this time, density (*ρ*) means a certain density that can maintain elasticity to enable mechanical vibration to alternate transverse and longitudinal waves in the transducer.

The analyzed gelatin, now in rubber form, yielded acoustic wave velocity (*c_g_*) and acoustic impedance (*Z_g_*) values of 3453 m/s and 4.13 MRay1, respectively, as shown in Equations (4) and (5).
(4)$$c_{T}\left({=c}_{g}\right)=\sqrt{\frac{E}{\rho }}$$
(5)$$Z_{g}=\rho c=\sqrt{E\rho }$$

Acoustic velocity represents the speed of ultrasound wave propagation through tissues, while acoustic impedance is a physical property of tissue indicating the amount of resistance encountered by an ultrasound beam. The ultrasound transmission velocity (*c_T_*) and attenuation coefficient (Att) of the probe were determined as 1544 m/s and 0.2 dB/cm·MHz [31], respectively. Here, acoustic wave velocity refers to the inherent speed at which ultrasonic waves pass from the transducer to the phantom [32]. At this time, the transmission space of ultrasonic waves to pass through the transducer and the phantom must match each other, and this definition can be expressed as acoustic impedance. Therefore, it is important that ultrasonic waves are perfectly transmitted from the transducer to the pantor; but at this time, some loss of ultrasonic waves may occur due to differences in the characteristics of the medium (lattice, density, elastic modulus, and characteristic impedance difference) [32]. At this time, the lower the ultrasonic transmission loss, the better, and the measure of minimizing ultrasonic transmission loss can be defined as the attenuation coefficient. In acoustic wave velocity, the actual transmission velocity, including elastic modulus (*E*), density (*ρ*), acoustic impedance, and attenuation coefficient, can be defined as ultrasound transmission velocity.

In this case, the impedance of the ultrasound probe and the proposed gelatin material must be matched for the ultrasound to be effectively transmitted to the target. Therefore, as per Equation (6) and as shown in Figure 10, the matched impedance (*Z_M_*) was calculated to be 11.13 MRay1 [33].
(6)$$Z_{M}=\sqrt{Z_{T}Z_{g}}$$

Impedance matching is crucial for preventing air penetration between the probe and the skin, as shown in Figure 11. This ensures that the acoustic transmission coefficient (*T*), as per Equation (7), is optimized. In this context, *T* should be close to 1 for optimal performance, and the analyzed value for *T* was determined to be 1.76 [33,34].
(7)$$T=1+\frac{Z_{g}-Z_{T}}{Z_{g}+Z_{T}}$$

Therefore, the power generated by the probe (PT), as follows the Equations (8) and (9), should be maximized as it passes through the gelatin material, reaching the target, as shown in Figure 12. Consequently, the efficiency (η = 100%) concerning power should be elevated to approach 100% [35]. In this context, *P_T_*, *P_g_*, and efficiency (*η*) were determined as 2.29 W, 2.76 mW, and 82.8%, respectively, as shown in Equation (10). Here, PDC represents the inherent power generated by the ultrasound system, driven by the supplied voltage (3V) of the ultrasound system.
(8)$$P_{T}=V_{T}I_{T}+V_{g}I_{g}=V_{T}\frac{I_{T}}{\pi }+V_{g}\frac{I_{g}}{\pi }$$
(9)$$P_{g}=\frac{1}{2}\left[\frac{I_{T}}{2}V_{T}\sqrt{2}cos\left(\frac{\pi }{4}\right)\right]+\frac{1}{2}\left[\frac{I_{g}}{2}V_{g}\sqrt{2}cos\left(\frac{\pi }{4}\right)\right]$$
(10)$$\eta =\frac{P_{T}-P_{g}}{P_{DC}}$$
where *V_T_*, *I_T_*, and *P_DC_* are 3V, 2.6 A, and 27.66 mA, respectively, while *V_g_* and *I_g_* are 3V and 0.726 μA [23,35]. If the efficiency is low, the power is dissipated as heat, generating heat at the transducer. Moreover, if air penetrates between the transducer and the skin, as shown in Figure 13, impedance mismatching can hinder power transmission, resulting in low efficiency and heat generation [35,36,37,38,39]. For therapeutic ultrasound, as depicted in Figure 13, the reflectance should be close to 0 (*R* = 0). However, in diagnostic ultrasound, where monitoring is through reflected images, higher reflectance, ideally above 0.5 (*R* > 0.5) as per Equation (11), indicates better performance [40]. Here, cg and d represent the acoustic velocity of a probe and the thickness of a gelatin layer, given as 3453 m/s and 1.0 cm, respectively [39]. Therefore, the minimum required reflectance (*R*) for monitoring should be above 50%. The reflectance analyzed through Equation (11) is 57.5%.
(11)$$R=\frac{Z_{g}^{2}-Z_{T}^{2}}{Z_{g}^{2}+Z_{T}^{2}-2jZ_{T}Z_{g}cot\left(\frac{2\pi fd}{c_{g}}\right)}={\left(\frac{{Z_{T}-Z}_{g}}{{Z_{T}+Z}_{g}}\right)}^{2}$$

This experiment, conducted using a soft solid gel, did not require approval from the Institutional Review Board (IRB) for human experiments. It is crucial to use the proposed soft solid gel to avoid the temperature impact on the ultrasound probe. Traditional gels tend to dry out after approximately 15 min, leading to a degradation in image quality during diagnostic evaluations [36,37,38,39,40,41,42,43,44], although there are existing research results that indicate images come out better in obese patients [45]. The experimental results, as shown in Table 2, indicate that the gel began to dry out around 15 min. Beyond this time frame, additional gel application would be inconvenient. However, if the proposed soft solid gel was used, the probe was not affected by heat and the gel could be used for a long time. Therefore, the gel achieved clear image quality regardless of environmental conditions.

The potential safety considerations for applying the soft solid gel to the human body must be taken into account. Recommendations for continued use include reproducibility and robustness. It is especially effective to follow the recommendations when using it repeatedly rather than one-time use. Therefore, as shown in Figure 14, if you want to reuse the soft solid gel, ethanol disinfection is essential. To test this reproducibility and stability, gels can be classified by time and image tests can be performed, as shown in Figure 14 and Table 2. Here, the process of soaking and removing the gel in ethanol was repeated at different time intervals, so that after ethanol disinfection, the gel could be reused without problems to obtain diagnostic quality images.

Thus, the tests were conducted for image evaluations, as given by Table 2 and Figure 14. Over a total duration of 72 h, tests were performed by storing the soft solid gel in alcohol for 6 h, 24 h, and 72 h. The results indicated that the shape of the soft solid gel remained unchanged over the 72 h period. However, upon inspecting the soft solid gel stored in alcohol for one week, melting of the gel was observed. The superior performance of the soft solid gel was shown in this experiment compared to that of traditional ultrasound gels.

Figure 15a shows ultrasonic gel testing at various time intervals using the ATS-539 phantom. Figure 15b shows the sharpness of the image of the manufactured gelatin soft gel after storing it in alcohol and performing a sharpness test at 1 h intervals for 72 h. The results are presented in Figure 15b. In this paper, the experiment using a soft solid gel did not require IRB agreement. This research on the proposed gelatin soft solid gel showed its functionality to be better than that of ultrasonic gel, including on grayscale, echo, and color. This means that it can overcome the tissue density dead zone to increase high resolution performance. To help imaging accuracy, it is proven that gelatin soft gel, especially the soft solid gel, can replace ultrasonic gel to simplify disinfection and ensure permanent use. Image corrosion testing of gelatin soft solid gel for 30 min is shown in Figure 16a. Ultrasonic gels were tested at different times using an ATS-539 phantom, where the gel was soaked in ethanol for 72 h, as shown in Figure 16b. Afterwards, the gel’s condition was tested again. Finally, Figure 16c measures the imaging resolution of the gelatin soft tissue gel.

In this context, the material must be used for the gelatin powder due to higher thermal conductivity compared to that of conventional gel material [27].

In addition, the ultrasonic gel ensures effective transmission of ultrasound waves and good contact with the skin of the subject under examination. The gel must have sufficient acoustic properties with optimal sound speed and characteristic impedance matching for efficient ultrasonic transmission and reception. It should be non-toxic for specific applications in the human body and enhance tissue visualization in ultrasound imaging. Based on a video comparison evaluation between the proposed soft solid gel and the conventional soft solid gel, the proposed soft solid gel exhibited excellent image quality [41,42].

The reason for using the ATS-539 phantom in video assessment is that this soft solid gel is designed as a standardized model capable of simulating various conditions. It allows for the reproduction of real patient conditions and enables the evaluation of ultrasound system performance in a realistic scenario. The ATS-539 enables repeatable experiments to obtain accurate results. It facilitates conducting experiments in a controlled environment and under consistent conditions, allowing for the assessment of the reliability and reproducibility of the generated images. The ATS-539 provides capabilities to evaluate various aspects of ultrasound examination. For example, it can provide experimental results for factors such as dead zone, ultrasonic arrival time, echo distance (deep), axial and lateral resolution, and grayscale. The ATS-539 is specialized for ultrasound testing and allows for the evaluation of various assessment parameters related to conditions. This helps in assessing the accuracy and performance of ultrasound examination and aids in improvement efforts. Therefore, the ATS-539 phantom is a standardized model that allows for the evaluation of reliability, reproducibility, and various assessment parameters.

Several limitations of solid soft gels must be considered. When storing solid soft gels in the refrigerator, a temperature of 1 to 3 °C must be maintained. The recommended storage period is 9 days at 80% humidity. Although the corrosion risk is low, extended refrigerated storage may affect quality, so use for only 9 days is suggested as a limitation. Short-term storage at room temperature (about 20 to 25 °C, about 12 days) requires special attention as cracking of the surface may occur. Using airtight containers is a great idea, but if possible, choose refrigerated storage. Another method is to avoid direct sunlight and store in a cool, well-ventilated place. If the storage method is incorrect, it begins to rot and causes a bad smell. In general, gelatin ingredients can be used for up to 12 days with proper care. One thing to note is that gelatin ingredients dissolve in hot water (above 36 °C) [46]. Using the traditional ultrasound gel for examinations requires continuous reapplication every 15 min as the gel tends to dry. However, the newly developed soft solid gel remains usable even after 15 min, providing sustained efficiency and economic benefits [47].

Gelatin, with tissue-mimicking characteristics, stands out for its flexibility, cost-effectiveness, and adjustability. Agar is sensitive to moisture, while silicon comes with complexity and high costs, and water requires meticulous management. While the general gel is used once and needs to be applied again after 15 min, the proposed method is considered economical in terms of unit cost because it can be stored for a long time and used multiple times after disinfection [47].

As shown in Figure 16, the individual values of the ultrasonic gel for grayscale, dead zone, vertical area, and horizontal area are 93.79 mm, 45.32 mm, 103.13 mm, and 83.86 mm, respectively. In contrast, the values for the soft solid gel are 105.64 mm in the grayscale, 34.48 mm in the dead zone, 141.0 mm in the vertical region, and 102.89 mm in the horizontal region. Therefore, the grayscale increased by 1.1 times, the dead zone remained the same, and the vertical zone increased by 1.2 times. As shown in Equation (12), the time value (T) in the horizontal zone was calculated by dividing that of the soft solid gel (*x*) by that of the American gel (*y*), resulting in a 1.2-fold increase. This calculation was performed by dividing the soft solid gel value by the corresponding ultrasonic gel value [25,26].
(12)$$T=\frac{y}{x}$$

When using echocardiography in actual clinical settings, there are many difficulties in obtaining diagnostic images. In particular, muscular patients have severe curvature and the probe cannot fully adhere to the skin, so excellent images could not be expected. However, obese patients have soft skin, so the probe adheres well and excellent images can be expected. Therefore, the proposed soft solid gel is expected to be able to increase the ultrasound projection rate by filling the area exposed to the air with solid gel when the probe is not properly adhered due to bending in a patient with large muscles. This paper obtained experimental results using a 2D imaging ultrasound probe. Generally, in clinical settings, both 2D imaging ultrasound and 3D imaging ultrasound use the same gel. At this time, the gel used in 2D imaging ultrasound can monitor excellent images in 3D imaging ultrasound. Therefore, since the proposed soft solid gel achieved excellent results in 2D imaging ultrasound, it is expected that it will also be able to obtain excellent results in 3D imaging ultrasound. Although 2D imaging ultrasound results were obtained due to limitations in the current research environment and system requirements, there is a need to attempt to obtain 3D imaging ultrasound results if a 3D imaging ultrasound system is secured in the future. In addition, there is a need to confirm superiority through experimental results of the proposed soft solid gel using 2D and 3D imaging ultrasound, various types of probes, and multiple modes of ultrasound.

## 5. Conclusions

In conclusion, the research findings demonstrate the superior imaging quality of gelatin ultrasound soft solid gels compared to conventional counterparts such as silicone soft solid gels, water soft solid gels, and commercially available ultrasound gels. Even after a 15 min video evaluation, the soft solid gel remained undamaged, contrasting with the tendency of ultrasound gel to dry out. The advantages of long-term usability and ease of production make soft solid gels a promising choice. Emphasizing the importance of meticulous mixing and precise temperature control during the manufacturing process, healthcare professionals and researchers handling soft solid gels should receive education on hygienic procedures and management methods. In this work, the experiments were conducted solely using a sector probe. For a more comprehensive evaluation, it is believed that this work requires additional research to increase objectivity by utilizing various probes, including linear and convex probes.

Adherence to the expiration date, proper storage conditions, and exclusive use to prevent cross-contamination are crucial. Upon commercialization, gelatin ultrasound soft solid gels are expected to play a significant role in disease diagnosis across various medical departments utilizing ultrasound for patient examinations.

## Figures and Tables

**Figure 1 diagnostics-14-00335-f001:**
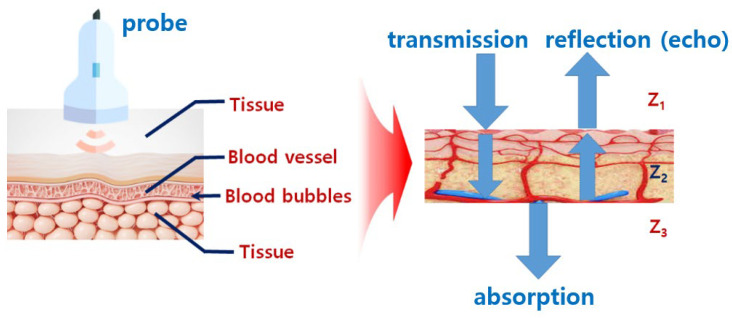
Ultrasound probe soft solid gel transmission and impedance-matching relationship diagram.

**Figure 2 diagnostics-14-00335-f002:**
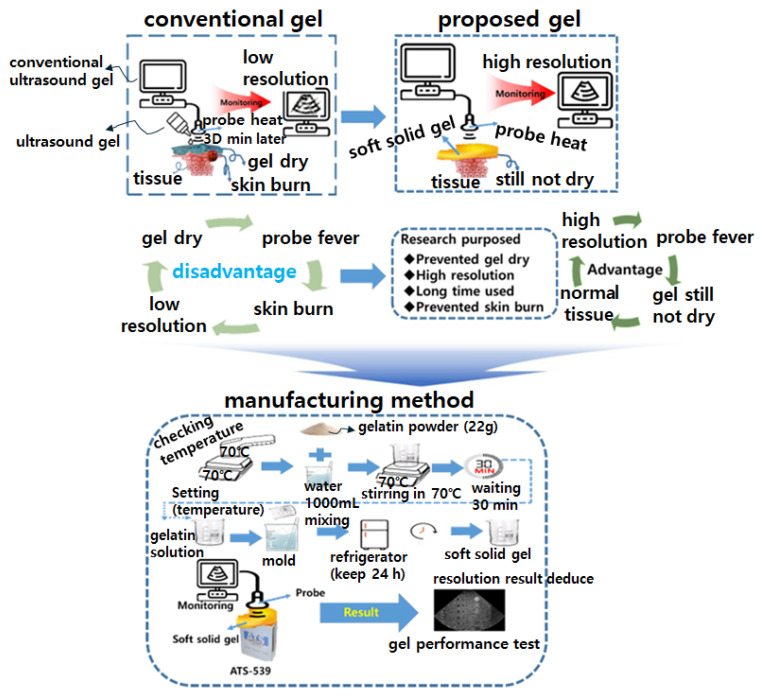
Manufacturing progress for proposed gelatin and performance test of the materials.

**Figure 3 diagnostics-14-00335-f003:**
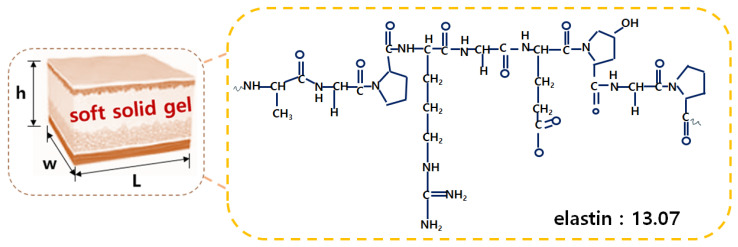
Structure and dimensions of the suggested gelatin.

**Figure 4 diagnostics-14-00335-f004:**
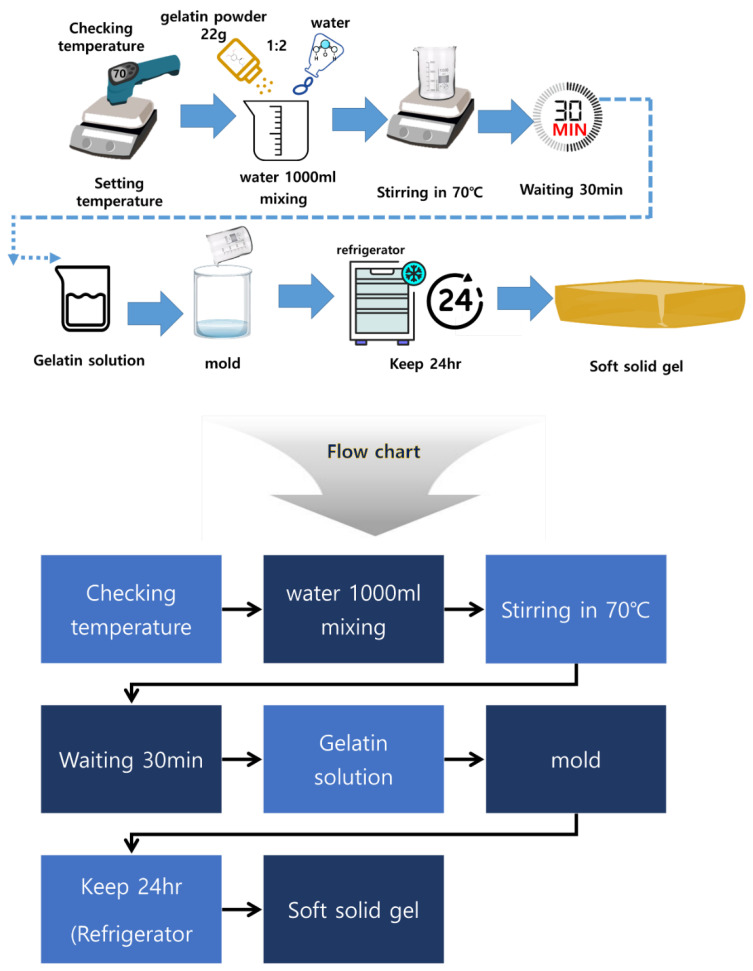
Manufacturing process of the gelatin-based soft solid gel.

**Figure 5 diagnostics-14-00335-f005:**
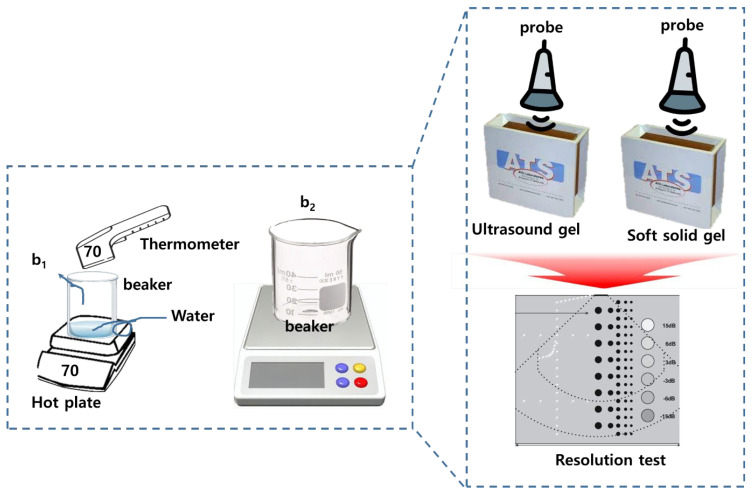
Ultrasonic imaging testing and evaluation of the soft solid gel (ATS-539).

**Figure 6 diagnostics-14-00335-f006:**
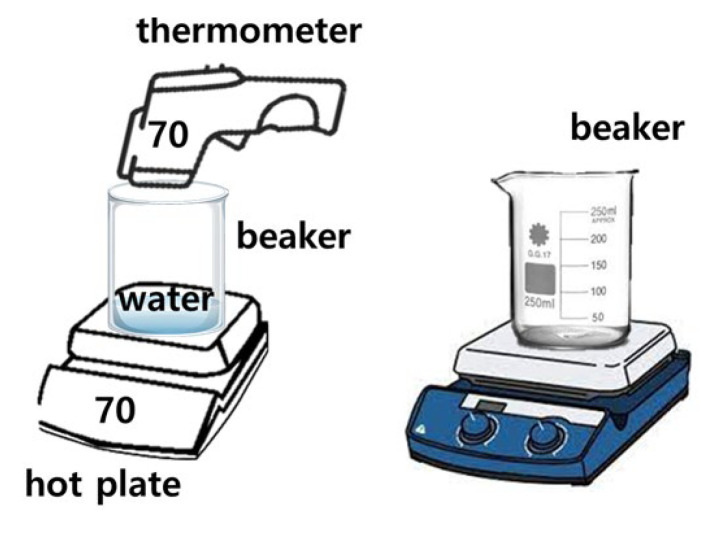
Manufactured construction of an ultrasound soft solid gel.

**Figure 7 diagnostics-14-00335-f007:**
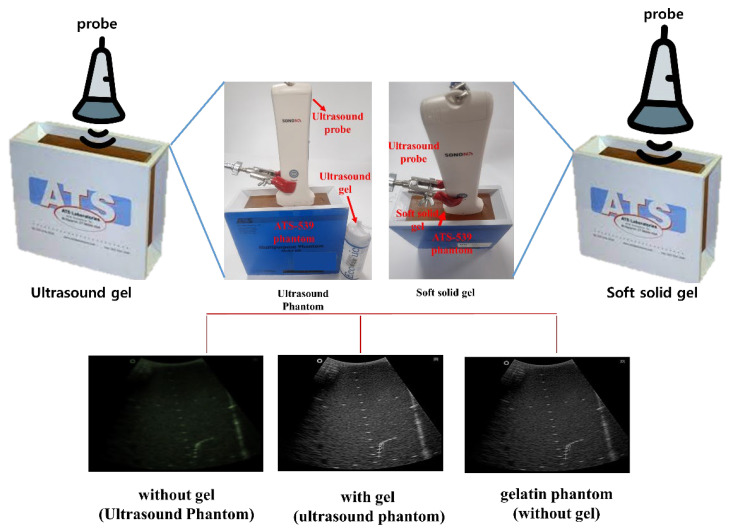
Imaging test with ultrasonic evaluation using ultrasonic gel (ATS-539) and a soft solid gel.

**Figure 8 diagnostics-14-00335-f008:**
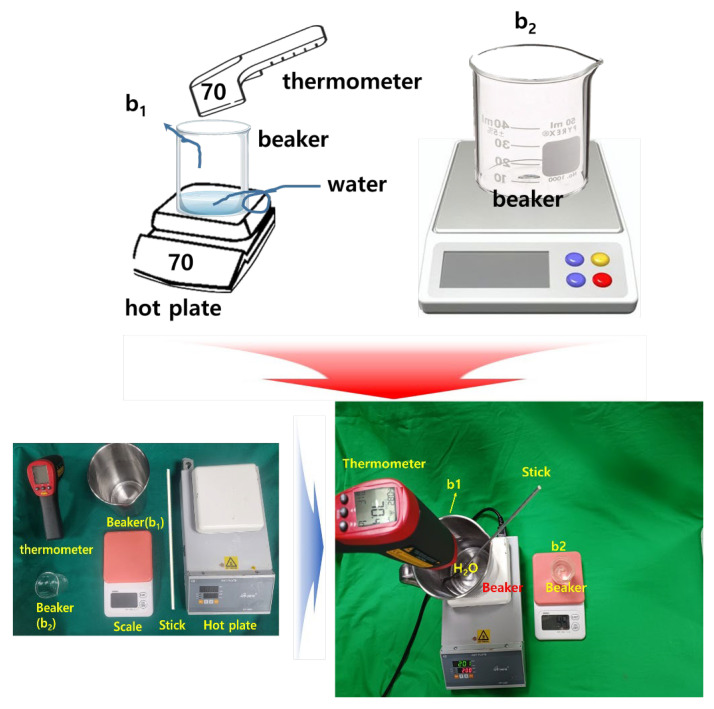
Process of measurement gelatin soft solid gel.

**Figure 9 diagnostics-14-00335-f009:**
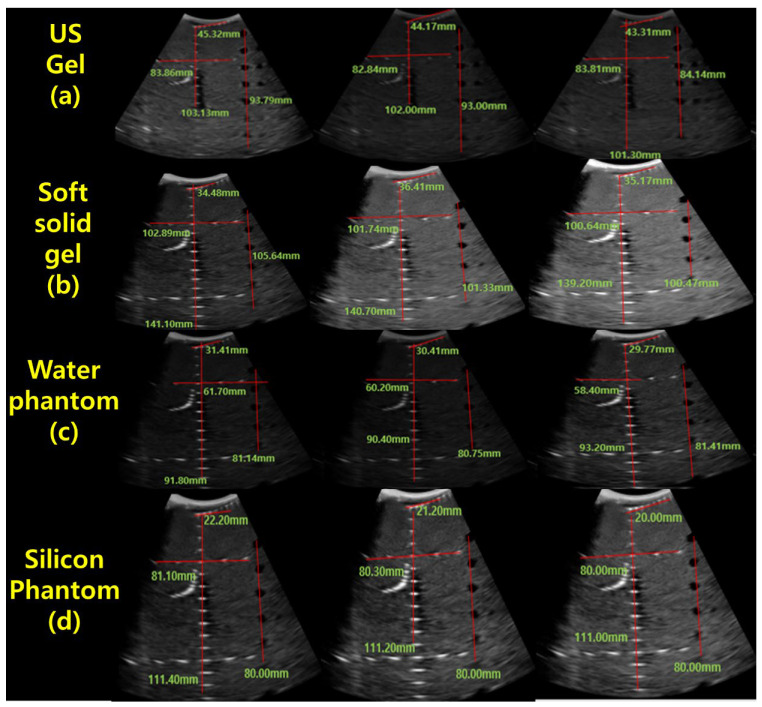
Ultrasonic imaging test evaluation of soft solid gel (ATS-539): (**a**) ultrasonic gel, (**b**) soft solid gel, (**c**) water phantom, and (**d**) silicon phantom.

**Figure 10 diagnostics-14-00335-f010:**
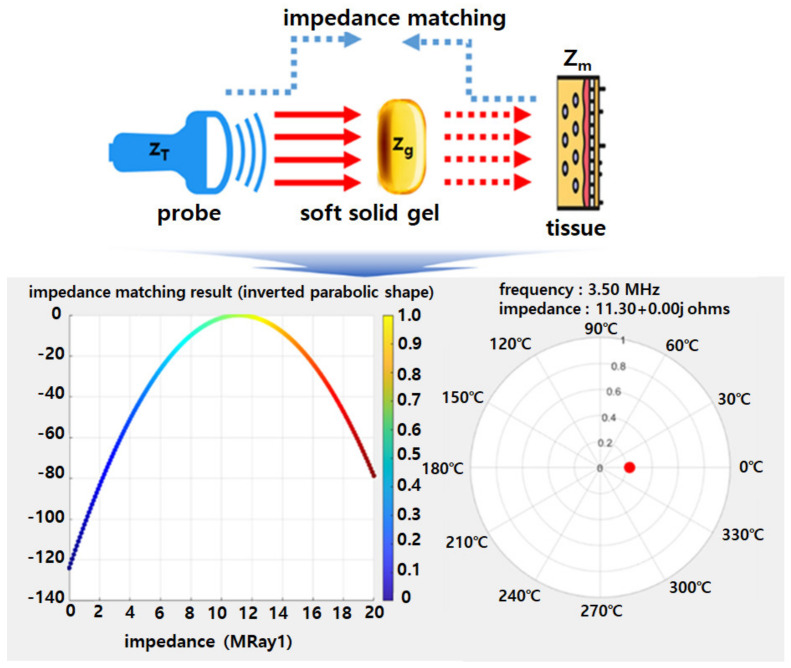
Simulation results for characteristic impedance of the probe and soft solid gel (marker of red color: impedance matching point).

**Figure 11 diagnostics-14-00335-f011:**
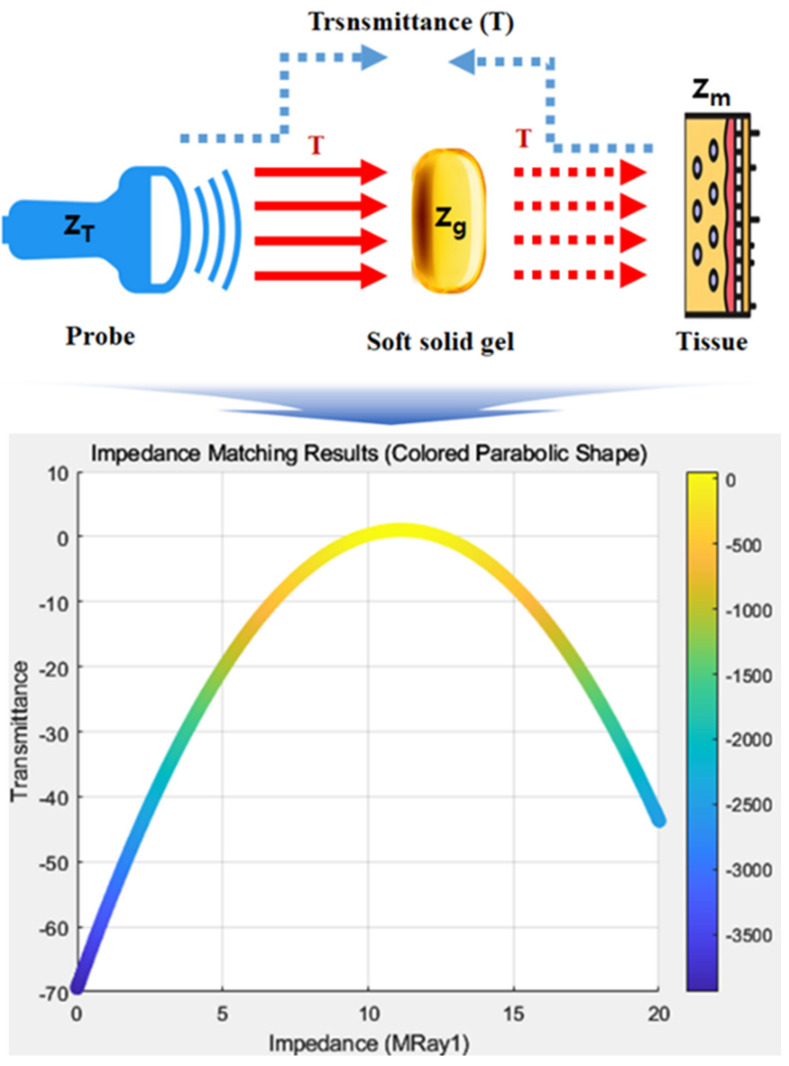
Simulation results for acoustic transmission coefficient (T).

**Figure 12 diagnostics-14-00335-f012:**
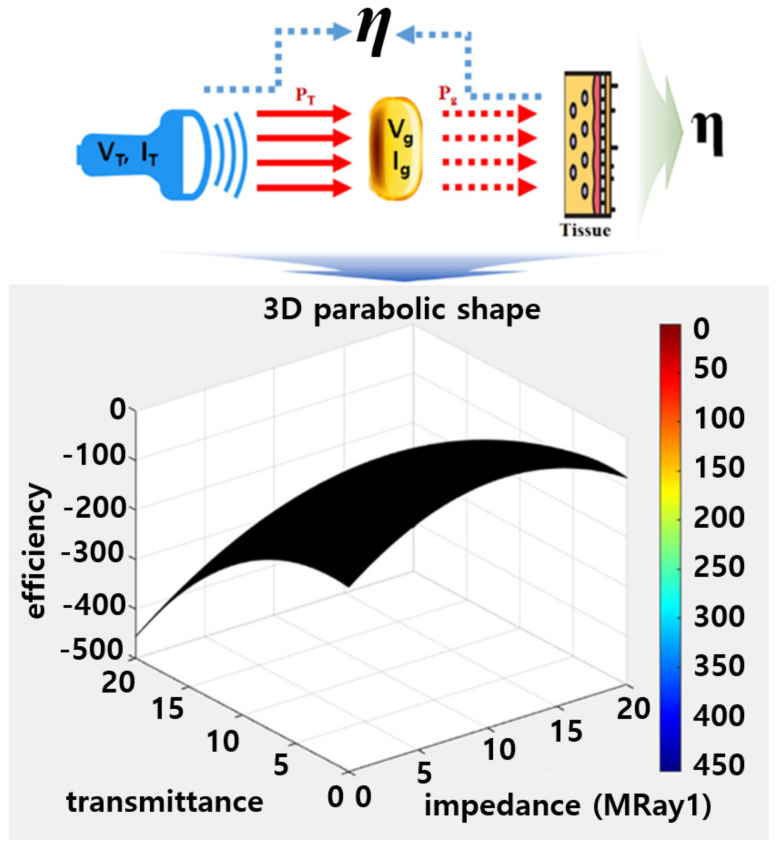
Simulation results for the transmission power and efficiency of the ultrasonic probe with soft solid gel.

**Figure 13 diagnostics-14-00335-f013:**
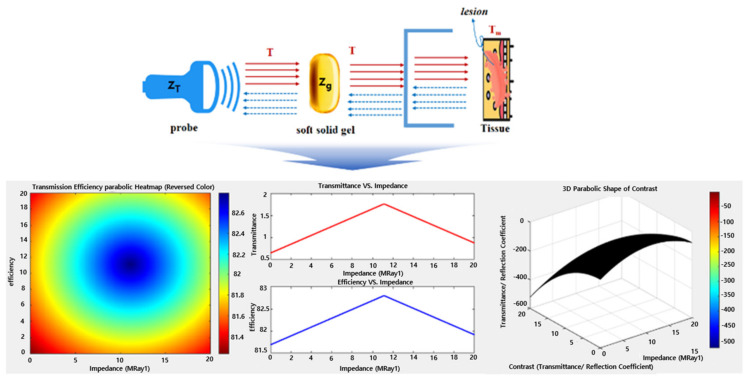
Stimulation results for the heat distribution, relative ultrasound reflectance, and impedance efficiency.

**Figure 14 diagnostics-14-00335-f014:**
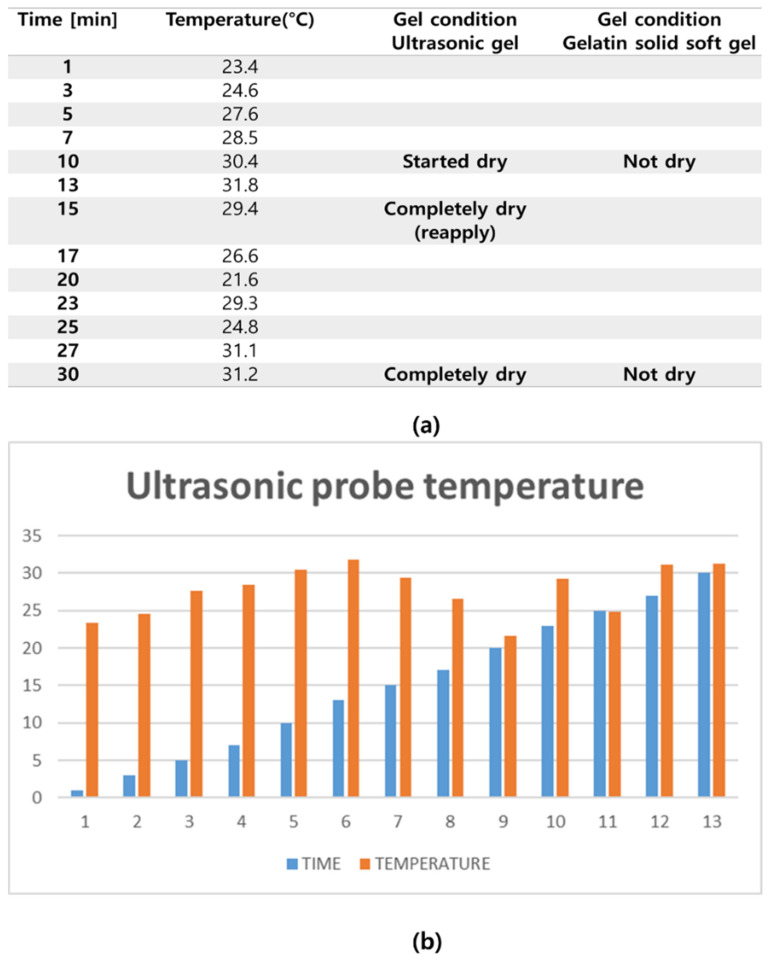
Analysis of probe temperature: (**a**) ultrasonic probe temperature test; (**b**) probe test chart.

**Figure 15 diagnostics-14-00335-f015:**
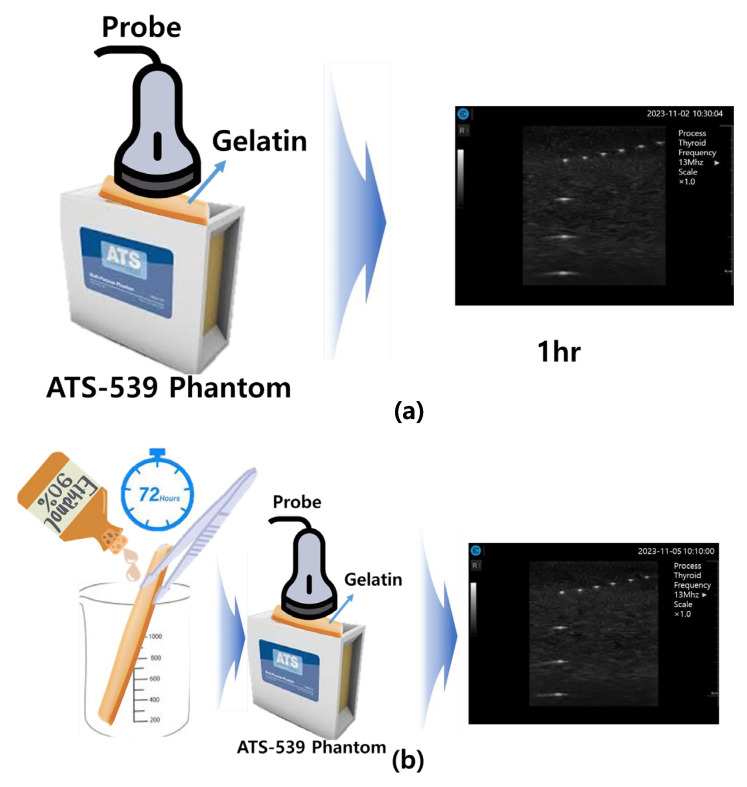
(ATS-539) phantom imaging resolution test (**a**) without alcohol included image (**b**) with alcohol included image.

**Figure 16 diagnostics-14-00335-f016:**
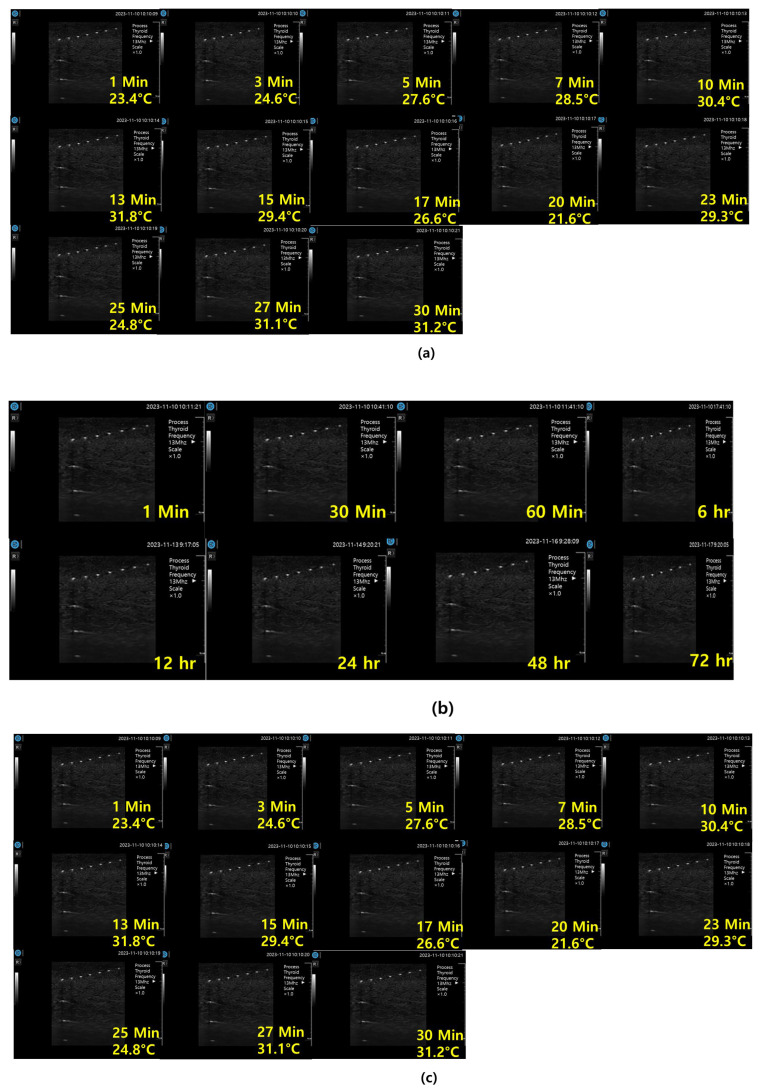
ATS-539 imaging test: (**a**) ultrasonic gel; (**b**) gelatin corrosion test; (**c**) gelation soft gel test.

**Table 1 diagnostics-14-00335-t001:** Comparison of the proposed gel and others.

Type	Gray Scale [mm]			Dead Zone [mm]			Vertical Zone [mm]			Horizontal Zone [mm]		
Content	AV *	±SD **	ERROR ***	AV	±SD	ERROR	AV	±SD	ERROR	AV	±SD	ERROR
US gel	87.0	4.8	2.4	44.3	1.01	0.58	102.1	0.92	0.5	83.5	0.65	0.03
Soft solid gel	102.4	2.3	1.1	35.4	0.98	0.56	140.3	1.0	0.6	101.8	1.1	0.7
Water soft solid gel	80.9	0.2	0.1	30.5	0.83	0.48	91.8	1.4	0.8	60.1	1.7	1.0
Silicon soft solid gel	80.0	0.01	0.01	21.1	1.1	0.64	111.2	0.2	0.1	80.5	0.6	0.3

* AV (average), ** SD (standard deviation), *** ERROR (standard error).

**Table 2 diagnostics-14-00335-t002:** Gelatin soft solid gel hourly test table.

Time [min/h]	Image	Significant
1 min	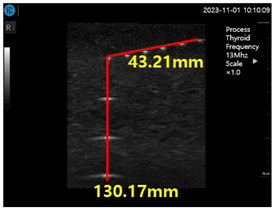	
30 min	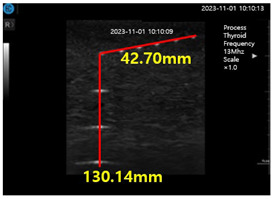	Not dry
60 min	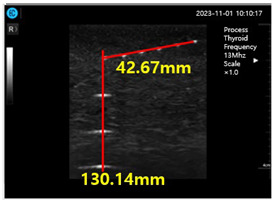	
6 h	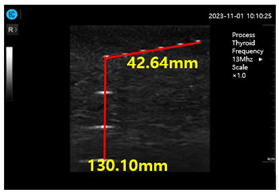	Not dry
12 h	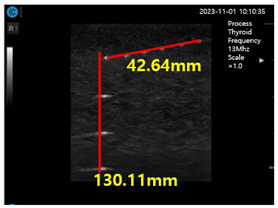	
24 h	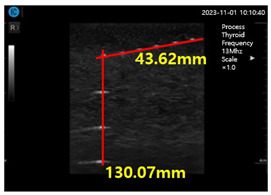	Not dry
48 h	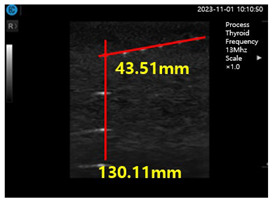	
72 h	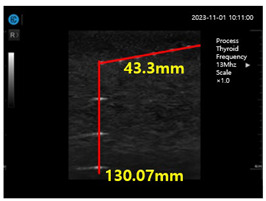	Not dry

## Data Availability

The data presented in this study are available upon request from the corresponding author. The data are not publicly available because of privacy and ethical restrictions.

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
