# Peer review of "Development of an Artificial Soft Solid Gel Using Gelatin Material for High-Quality Ultrasound Diagnosis"

_diagnostics, 2024, doi:10.3390/diagnostics14030335_

Round 1

Reviewer 1 Report

Comments and Suggestions for Authors

I congratulate the authors for the work carried out, for the meticulousness with which the gel construction process was described and for the focus on the chemical-physical characteristics of the ultrasound medium.

Abstract: ok

Intorduction: it is correct to discuss why gel is used for ultrasound and to report some evidence, however to say that one can get burned in a common ultrasound examination for diagnostic purposes, with the normal instruments and the normal energies used, seems a bit misleading and excessive. On the contrary, I really appreciated the second part of the introduction where it focuses on physical aspects of the interaction between gel, water and ultrasound

Manufacturing analysis and methods: well written

Results: line 202 please correct the visualization of the reference, maybe [29-32] is more correct than [29]-[32].

Discussion: well eritten

Conclusions: attend to the discussion and results

Figures: abundant and adequate for the description of a basic study on materials

Author Response

Comments 1

I congratulate the authors for the work carried out, for the meticulousness with which the gel construction process was described and for the focus on the chemical-physical characteristics of the ultrasound medium.

Answers 1

Thank you very much for your positive review of the paper. We will do our best to respond to your comments.

Comments 2

Abstract: ok

Intorduction: it is correct to discuss why gel is used for ultrasound and to report some evidence, however to say that one can get burned in a common ultrasound examination for diagnostic purposes, with the normal instruments and the normal energies used, seems a bit misleading and excessive. On the contrary, I really appreciated the second part of the introduction where it focuses on physical aspects of the interaction between gel, water and ultrasound.

Answers 2

Thank you for your advice and positive comments.

The explanation about burns was deleted because it could be misleading.

Comments 3

Manufacturing analysis and methods: well written

Answers 3

thank you I will review it as a whole and try to supplement it myself.

Comments 4

Results: line 202 please correct the visualization of the reference, maybe [29-32] is more correct than [29]-[32].

Answers 4

It's corrected. Please refer to the lines of 204 (sky blue) in the discussion.

Comments 5

Discussion: well eritten

Answers 5

thank you I will review it as a whole and try to supplement it myself.

Comments 6

Conclusions: attend to the discussion and results

Answers 6

thank you I will review it as a whole and try to supplement it myself.

Comments 7

Figures: abundant and adequate for the description of a basic study on materials

Answers 7

thank you I will review it as a whole and try to supplement it myself.

Reviewer 2 Report

Comments and Suggestions for Authors

This is an interesting paper, but it is poorly organized, and is not acceptable for publication in the present form.

Major points

1)    English: To be revised. If not, I cannot understand the detail.

2)    Structure: Poorly organized. Too long. (e.g. Don’t insert many explanations into the Results section. The explanations should be shifted into Discussion section). Please avoid repetitions as possible.

3)    Please describe precisely the acoustic characteristics of this solid gel: acoustic velocity, attenuation, density. ---.

Minor points

1)    Please mention the limitations of this gel soft solid gel.

2)    Is it expensive? Please mention its cost effectiveness.

3)    Video efficiency: I cannot understand what it means.

Comments on the Quality of English Language

To be revised. If not, I cannot understand the detail.

Author Response

Comments 1

This is an interesting paper, but it is poorly organized, and is not acceptable for publication in the present form.

Answers 1

Thank you very much for your positive review of the paper. However, I will do my best to respond and make corrections to your comments.

Comments 2

Major points

English: To be revised. If not, I cannot understand the detail.

Answers 2

We will do our best to revise the English sentences with the help of experts before they are published. thank you

Comments 3

Structure: Poorly organized. Too long. (e.g. Don’t insert many explanations into the Results section. The explanations should be shifted into Discussion section). Please avoid repetitions as possible.

Answers 3

I agree with your opinion because the long explanations in the results are boring for the readers. Therefore, the explanation and simulation in the results were judged as supplementary explanations and moved to the discussion session. Additionally, reflecting Reviewer 3's opinion, we shortened the long explanation in the conclusion and moved it to discussion. Thank you for your profound advice.

Comments 4

Please describe precisely the acoustic characteristics of this solid gel: acoustic velocity, attenuation, density. ---.

Answers 4

The meaning explanation of the characteristics was recorded in the lines of 204 to 208 and lines of 218 to 229 (pink) in the discussion session.

Comments 5

Minor points

Please mention the limitations of this gel soft solid gel.

Answers 5

Please refer to line of 331 to 340 (purple) in the discussion. The explanation is as follows:

Lastly, the proposed soft solid gel should be stored in a dry and well-ventilated place, avoiding direct sunlight. If the storage method is incorrect, it begins to rot and causes a bad smell. In general, gelatin ingredients can be used for 5 years with proper care. One thing to note is that gelatin ingredients dissolve in hot water (above 36℃).

Comments 6

Is it expensive? Please mention its cost effectiveness.

Answers 6

Please refer to the lines of 350 to 352 (gray) in the discussion. The explanation is as follows:

General gels are used once and need to be used again after 15 minutes, but the proposed method is considered economical in terms of unit cost because it can be stored for a long time and used multiple times after disinfection.

Comments 7

Video efficiency: I cannot understand what it means.

Answers 7

I revised the imaging resolution instead of video efficiency. Please refer to lines of 21 (red color)

Comments 8

Comments on the Quality of English Language

To be revised. If not, I cannot understand the detail.

Answers 8

We will do our best to revise the English sentences with the help of experts before they are published. thank you.

Reviewer 3 Report

Comments and Suggestions for Authors

The authors compared the effectiveness of the conventional gel and the newly developed soft gel in USG examination. It was a study that will be very useful in clinical practice. I think that this manuscript will contribute to the literature. It may be published in the journal after some revisions.

1. The authors used a probe with the same feature. I wonder what the effectiveness of the soft gel might be in probes with different frequencies? Do the effectiveness of gels change between 2D and 3D probes? It would be appropriate to discuss these issues.

2. What are the storage conditions of the soft gel and can it cause any deformity on the probe?

3. It can be discussed what effect the gel may have on image quality, especially in obese patients.

4. Conclusion section should be shortened.

5. What are the limitations of the study?

Author Response

Comments 1

The authors compared the effectiveness of the conventional gel and the newly developed soft gel in USG examination. It was a study that will be very useful in clinical practice. I think that this manuscript will contribute to the literature. It may be published in the journal after some revisions.

Answers 1

Thank you for your interest and review of my manuscript. I have done my best to edit it based on your comments.

Comments 2

The authors used a probe with the same feature. I wonder what the effectiveness of the soft gel might be in probes with different frequencies? Do the effectiveness of gels change between 2D and 3D probes? It would be appropriate to discuss these issues.

Answers 2

Please refer to the lines of 370 to 379 (dark blue) in the review.

Comments 3

What are the storage conditions of the soft gel and can it cause any deformity on the probe?

Answers 3

Please refer to line of 331 to 340 (purple) in the discussion. The explanation is as follows:

Lastly, the proposed soft solid gel should be stored in a dry and well-ventilated place, avoiding direct sunlight. If the storage method is incorrect, it begins to rot and causes a bad smell. In general, gelatin ingredients can be used for 5 years with proper care. One thing to note is that gelatin ingredients dissolve in hot water (above 36℃).

Comments 4

It can be discussed what effect the gel may have on image quality, especially in obese patients.

Answers 4

Please refer to Lines of 364-370 (red) in the discussion. The reference description is as follows:

When using echocardiography in actual clinical settings, there were many difficulties in obtaining diagnostic images. In particular, muscular patients had severe curvature, so the probe did not fully adhere to the skin, making it difficult to expect excellent images. However, obese patients have soft skin, so the probe adheres well and excellent images can be expected. Therefore, the proposed soft solid gel is expected to be able to increase the ultrasound projection rate by filling the area exposed to the air with solid gel when the probe is not properly adhered due to bending in a patient with large muscles.

Comments 5

Conclusion section should be shortened.

Answers 5

Thanks for your advice. I shortened the conclusion statement and moved it to a discussion session.

Comments 6

What are the limitations of the study?

Answers 6

Please refer to the lines of 370 to 379 (dark blue) in the review.

Round 2

Reviewer 2 Report

Comments and Suggestions for Authors

This manuscript has been well revised. I think it is suitable for publication.